# PPTSER: A Plug-and-Play Tag-guided Method for Few-shot Semantic Entity Recognition on Visually-rich Documents

## Abstract

Visually-rich document information extraction (VIE) is a vital aspect of document understanding, wherein Semantic Entity Recognition (SER) plays a significant role. However, the study of few-shot SER on visually-rich documents remains largely unexplored despite its considerable potential for practical applications. To address this issue, we propose a simple yet effective **P**lug-and-**P**lay **T**ag-guided method for few-shot **S**emantic **E**ntity **R**ecognition (**PPTSER**) on visually-rich documents. PPTSER is a pluggable method building upon off-the-shelf multimodal pre-trained models. It leverages the semantics of the tags to guide the SER task. In essence, PPTSER reformulates SER into entity typing and span detection, handling both tasks simultaneously via cross-attention. Experimental results illustrate that PPTSER outperforms fine-tuning baseline and existing few-shot methods, especially in low-data regimes. With full training data, PPTSER achieves comparable or superior performance to fine-tuning baseline. Specifically, on the FUNSD benchmark, our method improves the performance of LayoutLMv3 in 1-shot, 3-shot and 5-shot scenarios by 15.61%, 2.13%, and 2.01%, respectively. On the XFUND-zh benchmark, it improves the performance of LayoutLMv3 by 3.73%, 6.16%, and 4.01%, respectively. Overall, PPTSER demonstrates promising generalizability, effectiveness, and plug-and-play nature for few-shot SER on visually-rich documents. The codes will be available.

## 1 Introduction

Information extraction from visually-rich documents (VIE) is a process that concentrates on extracting pertinent information from various sources such as scanned images, documents, and PDF files. It effectively leverages layout and visual cues to decode the content enclosed within these documents (Xu et al., 2020). As an important part of VIE, Semantic Entity Recognition (SER) aims to extract entity spans from the visually-rich document. SER has been hailed as a significant advancement in the realm of document intelligence, and it has found widespread applications in numerous sectors.

Historically, the development of SER heavily relied on heuristic algorithms (Simon et al., 1997; Schuster et al., 2013). However, the advent of multi-modal pre-trained models (Xu et al., 2020; Li et al., 2021c; Gu et al., 2021; Huang et al., 2022b; Yu et al., 2023) has ushered in a rapid evolution in SER methodologies. These models, pre-trained on a large corpus of scanned documents in a self-supervised manner, have significantly enhanced the comprehension ability of SER.

Despite the remarkable achievements of the multi-modal pre-trained models, they often rely on extensive data for fine-tuning. However, acquiring a large volume of well-annotated SER data poses significant challenges such as: **(1)** Acquiring such data necessitates substantial financial resources and time. Annotators are required to label a multitude of OCR detection boxes in the document, adhering to meticulously designed guidelines. Identification of content within a box and accurately assigning labels to them are also tedious tasks. **(2)** The availability of data is often restricted due to privacy concerns. In scenarios involving sensitive information, such as invoices and insurance documents, data accessibility is severely limited due to the confidential nature of this information.

Despite the scarce research (Cheng et al., 2020; Yao et al., 2021; Wang & Shang, 2022) on few-shot Semantic Entity Recognition for visually-rich documents (few-shot SER), results have shown limi-

Figure 1: **(a)** The illustration of the traditional fine-tuning method, where *Doc. Tok.* refers to *Document Tokens*. **(b)** The overview of our PPTSER method. PPTSER replaces the last self-attention block with an improved attention block, which has less modules and parameters. And it eliminates the need for an extra classifier layer compared to traditional fine-tuning.

tations in terms of generality and performance, and were limited to the specific application scenario. This paper, inspired by the comprehension capabilities of pre-trained models and the selective focus nature of the attention mechanism, introduces a novel approach called PPTSER, a **P**lug-and-**P**lay **T**ag-guided method for few-shot **S**emantic **E**ntity **R**ecognition on visually-rich documents. The underlying principle of PPTSER consists of two main components: **(1) Semantic Understanding and Alignment:** Words related to SER tags are used as a prompt and are concatenated with the document's text tokens. This combined input is then fed into a multi-modal pre-trained model. The motivation behind is that the pre-trained model is expected to understand the semantics of both the document tokens and the tag-related prompt, thereby bringing the hidden states of the tokens and tag-related words for a specific entity type closer together. **(2) A New Improved Attention Mechanism:** The attention weight obtained from the last attention block between the tag-related prompt and document tokens is directly used as the probability of tokens belonging to different tags. Different heads of the attention mechanism could identify different spans, which is perfectly suited for the SER task that deals with numerous entity spans. By fully exploiting the weighted focus nature of the attention mechanism, the model eliminates the value transform layer, feed-forward layer in the last attention block, and does not require a separate classifier layer compared to traditional fine-tuning methods (as depicted in Figure 1), As a result, the total parameter is reduced.

Extensive experiments are conducted to show the effectiveness of PPTSER on several commonly-used SER benchmarks, which cover multiple languages, under different few-shot and the full training settings, and with different mainstream multi-modal pre-trained models.

The main contributions of this paper can be summarized as follows:

- We introduce PPTSER, a simple yet powerful plug-and-play tag-guided approach for few-shot SER. To the best of our knowledge, we are the first to propose a pluggable method that has demonstrated effectiveness on various pre-trained models and languages.

- Through the efficient utilization of the built-in self-attention mechanism within the pre-trained model, our method demonstrate advantages in terms of parameters to some degree, as compared to the traditional fine-tuning approaches.

- Experimental results show the superiority of our method over the traditional fine-tuning approaches in both few-shot and full-training-set scenarios. Moreover, PPTSER outperforms existing few-shot SER methods, thereby highlighting its overall efficacy.

## 2 RELATED WORKS

**SER on Visually-rich Documents.** The majority of research on SER focused on neural network based methods. While some early works leveraged textual features (Chiu & Nichols, 2016), image features (Guo et al., 2019), or combined them with layout features (Yu et al., 2021; Wang et al., 2021a) to address this issue, the emergence of multi-modal pre-trained models has revolutionized SER. These models are jointly pre-trained on a large scale unlabeled document dataset with textual, layout and even visual cues, so they have potentials to better understand a structured document. LayoutLM (Xu et al., 2020) was the first to combine textual and positional features of OCR boxes during the pre-training stage. Later, LayoutLMv2 (Xu et al., 2021a) and LayoutLMv3 (Huang et al., 2022b) further integrated visual features into the pre-training process using different architectures. Moreover, Wang et al. (2022a) advanced the model architecture with a language-agnostic layout

transformer in their work, LiLT. Alongside the advancements in model structures, other works (Appalaraju et al., 2021; Li et al., 2021b;a; Hong et al., 2022; Luo et al., 2023) have focused on the various pre-training tasks to facilitate the fusion of textual, layout, and visual image features at pre-training stage. While these advancements have improved SER capabilities to some extent, their few-shot learning abilities are yet to be thoroughly explored and understood.

**Few-shot SER on Visually-rich Documents.** Unlike SER, few-shot SER is not fully explored yet. Cheng et al. (2020) proposed a solution inspired by graph-matching techniques (Zanfir & Sminchisescu, 2018). They represented documents as graphs, with each node corresponding to an OCR-scanned box. For an unseen document, the type for entities was determined by comparing the relationships in the graph of unseen document with those in the graphs of support documents. Yao et al. (2021) also adopt a graph-matching approach to tackle this challenge. However, the type of entities was determined based on the relationships in different forms with more complex solvers. Taking a different way, Wang & Shang (2022) introduced a novel labeling scheme for SER. They reshaped SER as a generative task, and used LayoutReader (Wang et al., 2021b) to generate SER labels by predicting the next token. Although these studies preliminary explored few-shot SER, they lacked generality and plug-and-play adaptability, and their performances in general scenes require further exploration and improvement.

**Few-shot NER in Plain Texts.** While few-shot SER on visually-rich documents has only seen limited exploration, there has been extensive research on few-shot Named Entity Recognition (NER) in plain texts (Wang et al., 2022b; Das et al., 2022; Ma et al., 2022a; Cheng et al., 2023). However, only a few of these studies have considered the scenario where only limited data in the target domain is available. Huang et al. (2022a) proposed using the NER tag as a prompt and employing contrastive learning to address this issue. On the other hand, Ma et al. (2022b) reformulated few-shot NER as a Language Modeling task and used the pre-trained Masked Language Model head to predict a word that is related to the entity type for each token in the text. These works solely relied on textual information to tackle the NER problem, making them potentially unsuitable for the SER task, which often involves visually-rich documents. Moreover, they primarily addressed NER in sentences with few entity spans, while SER aims to extract entities from an entity-rich document with multiple text lines. In addition, their performance on multiple languages remains unexplored.

## 3 METHOD

### 3.1 TASK FORMULATION

SER is usually formulated as a sequence labeling task. For given tokens from the document $x = [x_i], i = 1, 2, ..., n$, SER aims to assign a label $y_i \in \mathbb{C}$ for each token $x_i$, where $\mathbb{C}$ is the SER label space. Subsequently, entity spans would be analyzed from the labeled tokens according to a specific scheme, such as `BIO` (Ramshaw & Marcus, 1995) and `IO` (Tjong Kim Sang & De Meulder, 2003).

In this paper, we primarily focus on the **In-Label-Space** setting for few-shot SER. Specifically, the multi-modal model pre-trained is firstly fine-tuned on a small number of $M$ annotated documents, denoted as $\mathcal{D}^{train}$, with label space $\mathbb{C}$. After fine-tuning, the model is directly evaluated on a test set $\mathcal{D}^{test}$ with the same label space $\mathbb{C}$ as $\mathcal{D}^{train}$. This task presents a significant challenge as the model needs to learn the SER task with only a limited number of training samples.

It is crucial to note that in the context of few-shot SER, the few-shot setting of *N-way K-shot* indicates that *there are K documents containing entities of a specific type across N categories*, as visually-rich documents are annotated at document level. Furthermore, a single document often contains entity spans of distinct types, leading to potential overlaps between the support sets for different entity types across $N$ categories. Consequently, the number of annotated documents $M < N \times K$.

### 3.2 PPTSER

The fundamental concept and flow chart of PPTSER is shown in Figure 2. The method begins with the construction of a prompt based on SER tags. This prompt is then concatenated with the document tokens and jointly encoded using a unified pre-trained model. Within the transformer architecture of our model, attention weights between document tokens and the tag-related prompt are computed in hierarchical attention blocks. We use the attention weight between the tag-related

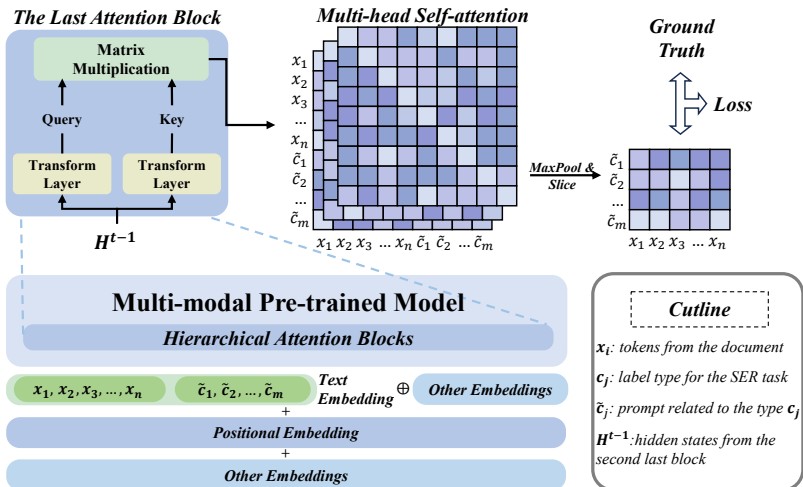

Figure 2: The overall architecture of PPTSER. The presence and format of *Other Embeddings* vary depending on the pre-trained model type. In this architecture, the tokens extracted from documents and the tag-related prompt are concatenated and subsequently encoded with the pre-trained model. The attention weight, obtained from the last attention block between tokens and the prompt, is then used to ascertain whether the tokens correspond to the respective label type.

prompt and document tokens, which can be considered as a form of cross-attention, obtained from the last attention block as the probability distribution of tokens belonging to different SER entity types. Finally, the loss between the cross-attention weight and the ground truth is calculated, which is used to train and optimize the model.

### 3.2.1 TAG-RELATED PROMPT CONSTRUCTION AND TARGET GENERATION

For an SER task with the label space $\mathbb{C}$, we need to construct tag-related words $\tilde{c}_i$ for each $c_i \in \mathbb{C}$, and then the tag-related prompt $\tilde{\mathbb{C}} = \{\tilde{c}_i\}, i = 1, 2, ...m$ is built. In PPTSER, we simply use the tag names as the tag-related words.

To ensure consistency with other traditional fine-tuning methods and enable PPTSER to accurately identify the boundaries of entity spans, we employ BIO tagging scheme in our method. However, when dealing with an SER task involving entity types $\mathbb{E} = \{e_i | e_0 = Other\}, i = 0, 1, 2, ..., m$ (where *Other* represents the entities that are not of interest), the label space would be $\mathbb{C} = \{e_0, B_{e_i}, I_{e_i}\}$, and the prompt would be $\tilde{\mathbb{C}} = \{e_0, beginning\, of\, e_i, inner\, of\, e_i\}$, where $i = 1, 2, ..., m$. In such scenario, the prompt $\tilde{\mathbb{C}}$ becomes not only semantically redundant but also excessively long, potentially impeding the effective semantic learning of the document tokens.

Therefore, we propose a reframing of the SER task with a BIO tagging scheme into two separate tasks: *entity typing* and *span detection*. Entity typing focuses on assigning an entity type for each token in the document, while span detection aims to identify whether tokens are at the beginning or interior of an entity span.

To further clarify, let's consider an SER task using BIO tagging scheme with a predefined entity type set $\mathbb{E} = \{e_i | e_0 = Other\}, i = 0, 1, 2, ..., m$. For entity typing, the label space and the tag-related prompt would be $\mathbb{C}^{ent.} = \{c_i^{ent.} | c_i^{ent.} = e_i\}$ and $\tilde{\mathbb{C}}^{ent.} = \{\tilde{c}_i^{ent.} | \tilde{c}_i^{ent.} = c_i^{ent.}\}$, where $i = 0, 1, 2, ..., m$; And for span detection, the label space and the prompt would be $\mathbb{C}^{det.} = \{c_1^{det.}, c_2^{det.}\}$ and $\tilde{\mathbb{C}}^{det.} = \{\tilde{c}_1^{det.}, \tilde{c}_2^{det.}\}$, where $\mathbb{C}^{det.} = \tilde{\mathbb{C}}^{det.} = \{beginning, inner\}$; Then the full label space and the full tag-related prompt would be $\mathbb{C} = \mathbb{C}^{ent.} \cup \mathbb{C}^{det.} = \{c_i^{ent.}, c_j^{det.}\}$ and $\tilde{\mathbb{C}} = \tilde{\mathbb{C}}^{ent.} \cup \tilde{\mathbb{C}}^{det.} = \{\tilde{c}_i^{ent.}, \tilde{c}_j^{det.}\}$, where $i = 0, 1, 2, ..., m; j = 1, 2$. For a token with an entity type of $e_i (i \neq 0)$ located at the $beginning/inner$ of an entity span, the corresponding labels would be $c_i$ for entity typing and $beginning/inner$ for span detection. However, for the token with an entity type of $Other$, we assign $-1$ as the span detection label, indicating that the specific location of it within an entity span is irrelevant and the loss for span detection is ignored here. Consequently, we can formulate the entity typing target $\boldsymbol{y}^{ent.} = [y_i^{ent.}]$ and the span detection target $\boldsymbol{y}^{det.} = [y_i^{det.}]$, where $i = 1, 2, ..., n$.

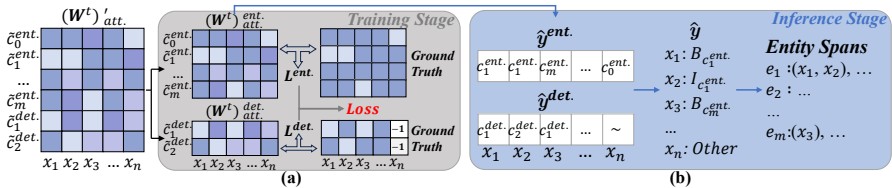

Figure 3: **(a)** PPTSER at training stage. Losses of entity typing and span detection are computed separately and then combined for the overall loss calculation. **(b)** PPTSER at inference stage. Grouped predictions of entity typing and span detection are utilized to analyse the entity spans.

It is worth emphasizing that while entity typing and span detection are distinct tasks, our PPTSER framework handles them simultaneously. And prompts for them $\tilde{\mathbb{C}}^{ent.}$ and $\tilde{\mathbb{C}}^{det.}$ are encoded in parallel, allowing them to benefit from each other during the learning process.

### 3.2.2 CROSS-ATTENTION WITHIN THE PRE-TRAINED MODEL

Once the tag-related prompt $\tilde{\mathbb{C}}$ is constructed, it is concatenated with the document tokens $\boldsymbol{x} = [x_i], i = 1, 2, ..., n$, forming the encoder input $\boldsymbol{x}' = \boldsymbol{x} \bigoplus \tilde{\mathbb{C}} = [x_i, \tilde{c}_j^{ent.}, \tilde{c}_k^{det.}], i = 1, 2, ..., n; j = 0, 1, 2, ..., m; k = 1, 2$. Then, $\boldsymbol{x}'$ is encoded in the pre-trained model. Let's denote the hidden states from the second last block as $\boldsymbol{H}^{t-1}$:

$$\boldsymbol{H}^{t-1} = [\boldsymbol{h}_i^{t-1}, \tilde{\boldsymbol{h}}_j^{t-1}, \tilde{\boldsymbol{h}}_k^{t-1}] \tag{1}$$

where $\boldsymbol{h}_i^{t-1}, \tilde{\boldsymbol{h}}_j^{t-1}, \tilde{\boldsymbol{h}}_k^{t-1}$ are the hidden states for $\boldsymbol{x}, \tilde{\mathbb{C}}^{ent.}, \tilde{\mathbb{C}}^{det.}$, correspondingly.

Then, $\boldsymbol{H}^{t-1}$ is partitioned into multiple segments $\boldsymbol{H}_i^{t-1}$ along the channel dimension, where queries $\boldsymbol{Q}_i^t$ as well as keys $\boldsymbol{K}_i^t$ of head $i$ are transformed. Here, $i = 1, 2, ..., l$ represents different heads for the attention mechanism. And the self-attention weight of different heads is computed as follows:

$$(\boldsymbol{W}_i^t)_{att.} = \boldsymbol{Q}_i^t (\boldsymbol{K}_i^t)^T \tag{2}$$

where $(\boldsymbol{W}_i^t)_{att.}$ is a matrix with the shape of $(n+m+3) \times (n+m+3)$. From this matrix, we extract a sub-matrix $(\boldsymbol{W}_i^t)'_{att.}$ that takes the prompt as queries and the document tokens as keys, which possesses the shape of $(m+3) \times n$. $(\boldsymbol{W}_i^t)'_{att.}$ can be viewed as a form of cross-attention within the self-attention, which depicts the relationship between the tag-related prompt and document tokens.

We hypothesize that distinct heads of the attention mechanism enable the prompt to focus on distinct entity spans, which is suitable for the entity-rich scenario in visually-rich documents. We select the maximum weight across heads to get a summary relationship between the prompt and tokens:

$$(\boldsymbol{W}^t)'_{att.} = \max_{i \in \{1,2,...,l\}} (\boldsymbol{W}_i^t)'_{att.} \tag{3}$$

Further, $(\boldsymbol{W}^t)'_{att.}$ is partitioned into two components, namely $(\boldsymbol{W}^t)_{att.}^{ent.}$ and $(\boldsymbol{W}^t)_{att.}^{det.}$ as shown in Figure 3(a). These components use the hidden states of $\tilde{\mathbb{C}}^{ent.}$ and $\tilde{\mathbb{C}}^{det.}$ as queries, and possess the shape of $(m+1) \times n$ and $2 \times n$, correspondingly. $(\boldsymbol{W}^t)_{att.}^{ent.}$ and $(\boldsymbol{W}^t)_{att.}^{det.}$ represent the probability distribution for document tokens belonging to distinct tags. The loss is then calculated as follows:

$$Loss = L^{ent.} + \alpha L^{det.}$$
$$= [-\frac{1}{n}\sum_{i=1}^{n} \frac{exp(w_{pi}^{ent.})}{\sum_{j=0}^{m} exp(w_{ji}^{ent.})}] + \alpha[-\frac{1}{n}\sum_{i=1}^{n} \frac{exp(w_{qi}^{det.})}{\sum_{j=1}^{2} exp(w_{ji}^{det.})}] \tag{4}$$

where $L^{ent.}$ and $L^{det.}$ are the losses for entity typing and span detection, $w_{ij}^{ent.}$ and $w_{ij}^{det.}$ are elements in $(\boldsymbol{W}^t)_{att.}^{ent.}$ and $(\boldsymbol{W}^t)_{att.}^{det.}$, and $y_i^{ent.} = c_p^{ent.}, y_i^{det.} = c_q^{det.}$. Besides, $\alpha$ is the ratio factor to balance the losses, and we set $\alpha = 0.1$ for models with segment-level positional embeddings and $\alpha = 1.5$ for models with word-level positional embeddings.

### 3.2.3 DECODING DURING THE INFERENCE STAGE

The inference stage is shown in Figure 3(b). We first apply the $argmax$ operation on $(\boldsymbol{W}^t)_{att.}^{ent.}$ and $(\boldsymbol{W}^t)_{att.}^{det.}$ along distinct prompt words to get the predicted tag with the highest probability:

$$\hat{y}_i^{ent.} = \underset{j \in \{0,1,2...,m\}}{argmax} w_{ji}^{ent.} \tag{5}$$

$$\hat{y}_i^{det.} = \underset{j \in \{1,2\}}{\operatorname{argmax}} \, w_{ji}^{det.} \tag{6}$$

Then the prediction with $\mathtt{BIO}$ tagging scheme $\hat{\boldsymbol{y}} = \{\hat{y}_i\}, i = 1, 2, ..., n$ is formulated as follows:

$$\hat{y}_i = \begin{cases} B_{\hat{y}_i^{ent.}} & , \hat{y}_i^{ent.} \neq Other, \hat{y}_i^{det.} = beginning \\ I_{\hat{y}_i^{ent.}} & , \hat{y}_i^{ent.} \neq Other, \hat{y}_i^{det.} = inner \\ Other & , \hat{y}_i^{ent.} = Other \end{cases} \tag{7}$$

Finally, the entity spans are analysed from $\hat{\boldsymbol{y}}$ according to the $\mathtt{BIO}$ tagging scheme. It's worth noting that, during this analysing, we assign the entity type $Other$ to those spans that do not adhere to the $\mathtt{BIO}$ tagging scheme, specifically those entity spans that begin with the token predicted as $\hat{y}_i^{det.} = inner$. This operation, aimed at enhancing prediction accuracy, is utilized across all methods we implemented, including PPTSER and the methods we used for a fair comparison.

## 4 EXPERIMENTS

### 4.1 EXPERIMENTAL SETTINGS

**Benchmarks.** We conducted experiments on several widely used SER benchmarks, including FUNSD (Jaume et al., 2019), CORD (Park et al., 2019) and XFUND (Xu et al., 2022). FUNSD is a benchmark specifically designed for form understanding, consisting of 199 noisy scanned documents related to market reports, commercials, and more. CORD focuses on receipt understanding and includes annotation at two levels: coarse-grained annotations such as *menu* and *total*, as well as fine-grained annotations like *menu.unitprice* and *menu.price*. This benchmark provides an official split of training, validation and test sets, and we strictly follow the procedure by selecting the model weight that achieved the best performance on the validation set for testing on the test set. XFUND is a document understanding benchmark covering multiple languages. In this article, our primary focus is on the Chinese subset of XFUND, denoted as XFUND-zh.

**Few-shot Settings.** We evaluated PPTSER on 1-shot, 3-shot, 5-shot, 7-shot and the full training set scenarios. Since the aforementioned benchmarks do not provide official divisions for few-shot scenarios, we established our own few-shot divisions following the procedure described in A.1. We aimed to select as few samples as possible while meeting the required sample numbers, which aligns with the real-world application. Due to the inherent instability of experiments with few-shot setting, we generated 5 divisions for each scenario using different random seeds, and we performed experiments on each division with 2 distinct random seeds. Hence, our experiment result is the average of 10 runs, ensuring the reliability and credibility of our findings.

### 4.2 COMPARISONS WITH EXISTING FINE-TUNING METHODS

**Setup.** The foundation for our method is built upon several widely used multi-modal pre-trained models, incorporating different combinations of modalities as input. This includes **BROS** (Hong et al., 2022) and **LiLT** (Wang et al., 2022a) with textual and layout input, and **LayoutLMv2** (Xu et al., 2021a) and **LayoutLMv3** (Huang et al., 2022b) with textual, layout and image input. Since BROS only supports English, we only tested it on FUNSD and CORD. For testing on XFUND-zh, we used **LayoutXLM** (Xu et al., 2021b), which is the multilingual version of LayoutLMv2.

**Results.** Table 1 showcases the results of PPTSER compared to traditional fine-tuning methods. The results clearly demonstrate that our PPTSER outperforms traditional fine-tuning methods across all tested scenarios and benchmarks. This underscores the superior performance of PPTSER in diverse language contexts with various base models.

Overall, both PPTSER and the fine-tuning method demonstrate improved performance as the training data increases. However, our PPTSER consistently outperforms previous fine-tuning methods across all few-shot settings, particularly when the training data is exceptionally scarce. In the 1-shot scenario on FUNSD, where only a single annotated document is available, PPTSER achieves the gains of **+6.31%** with BROS, **+3.04%** with LiLT, **+3.95%** with LayoutLMv2 and the highest gain of **+15.62%** with LayoutLMv3. This highlights the suitability of our PPTSER for few-shot scenarios. Furthermore, it's worth noting that even when trained with the full training data, our PPTSER

Table 1: F1 score (%) of PPTSER and traditional Fine-tuning methods. F1 score in **Bold** is better between our PPTSER and Fine-tuning. *FT* refers to *Fine-tuning* methods.

| Modality | | Text + Layout | | | | Text + Layout + Image | | | |
|---|---|---|---|---|---|---|---|---|---|
| Methodology | | BROS (AAAI 22) | | LiLT (ACL 22) | | LayoutLMv2 (ACL 21) | | LayoutLMv3 (MM 22) | |
| | | FT | **Ours** | FT | **Ours** | FT | **Ours** | FT | **Ours** |
| FUNSD | 1-shot | 48.08 | **54.39** $^{\uparrow 6.31}$ | 52.60 | **55.64** $^{\uparrow 3.04}$ | 48.22 | **52.17** $^{\uparrow 3.95}$ | 46.37 | **61.98** $^{\uparrow 15.61}$ |
| | 3-shot | 64.34 | **67.70** $^{\uparrow 3.36}$ | 67.64 | **69.17** $^{\uparrow 1.52}$ | 61.66 | **63.64** $^{\uparrow 1.98}$ | 74.73 | **76.86** $^{\uparrow 2.13}$ |
| | 5-shot | 67.77 | **70.64** $^{\uparrow 2.87}$ | 73.29 | **75.26** $^{\uparrow 1.97}$ | 65.86 | **67.49** $^{\uparrow 1.63}$ | 79.52 | **81.53** $^{\uparrow 2.01}$ |
| | 7-shot | 68.21 | **71.96** $^{\uparrow 3.75}$ | 73.39 | **75.71** $^{\uparrow 2.32}$ | 66.55 | **68.83** $^{\uparrow 2.28}$ | 79.84 | **81.60** $^{\uparrow 1.76}$ |
| | Full Data | 83.83 | **83.91** $^{\uparrow 0.08}$ | 88.95 | **89.07** $^{\uparrow 0.12}$ | 83.52 | **83.72** $^{\uparrow 0.20}$ | 91.15 | **92.01** $^{\uparrow 0.86}$ |
| CORD | 1-shot | 66.28 | **68.48** $^{\uparrow 2.20}$ | 70.04 | **75.57** $^{\uparrow 5.54}$ | 69.61 | **69.97** $^{\uparrow 0.36}$ | 70.35 | **74.19** $^{\uparrow 3.84}$ |
| | 3-shot | 79.02 | **79.61** $^{\uparrow 0.59}$ | 81.64 | **83.83** $^{\uparrow 2.19}$ | 80.63 | **81.66** $^{\uparrow 1.03}$ | 82.05 | **85.27** $^{\uparrow 3.22}$ |
| | 5-shot | 84.04 | **84.37** $^{\uparrow 0.34}$ | 85.52 | **87.06** $^{\uparrow 1.54}$ | 84.32 | **84.53** $^{\uparrow 0.21}$ | 85.83 | **87.77** $^{\uparrow 1.94}$ |
| | 7-shot | 83.68 | **84.09** $^{\uparrow 0.42}$ | 85.35 | **87.73** $^{\uparrow 2.38}$ | 84.76 | **85.31** $^{\uparrow 0.55}$ | 86.94 | **88.48** $^{\uparrow 1.54}$ |
| | Full Data | 95.72 | **95.75** $^{\uparrow 0.03}$ | 95.80 | **96.04** $^{\uparrow 0.25}$ | 95.20 | **95.63** $^{\uparrow 0.44}$ | 96.34 | **96.39** $^{\uparrow 0.05}$ |
| XFUND-zh | 1-shot | - | - | 60.10 | **67.64** $^{\uparrow 7.54}$ | 60.28 | **68.26** $^{\uparrow 7.98}$ | 52.92 | **56.65** $^{\uparrow 3.73}$ |
| | 3-shot | - | - | 72.61 | **74.17** $^{\uparrow 1.56}$ | 74.37 | **77.20** $^{\uparrow 2.83}$ | 69.08 | **75.24** $^{\uparrow 6.16}$ |
| | 5-shot | - | - | 77.40 | **79.40** $^{\uparrow 2.00}$ | 81.43 | **82.34** $^{\uparrow 0.91}$ | 75.25 | **79.26** $^{\uparrow 4.01}$ |
| | 7-shot | - | - | 80.47 | **81.38** $^{\uparrow 0.91}$ | 82.25 | **83.66** $^{\uparrow 1.41}$ | 77.85 | **80.97** $^{\uparrow 3.12}$ |
| | Full Data | - | - | 90.47 | **90.61** $^{\uparrow 0.14}$ | 90.25 | **90.79** $^{\uparrow 0.54}$ | 91.61 | **92.19** $^{\uparrow 0.58}$ |

still achieves comparable performance to the fine-tuning method, and even outperforms it in certain scenarios. For example, we observe a gain of **+0.86%** on FUNSD with LayoutLMv3. This full data setting is often overlooked in other few-shot research, further underscoring the superiority of our approach when dealing with varying amounts of available data. For more experimental results, please refer to A.4.1.

Our findings demonstrate that PPTSER is highly adaptive to different amounts of training data with distinct base models, making it an effective method for addressing the SER problem.

### 4.3 COMPARISONS WITH EXISTING FEW-SHOT METHODS

Table 2: F1 score (%) of PPTSER and other Few-shot methods. F1 score in **Bold** is the best, and that with underline is the second best.

| Modality | | Text | | Text + Layout | | | Text + Layout + Image | |
|---|---|---|---|---|---|---|---|---|
| Methodology | | EntLM (NAACL 22) | COPNER (COLING 22) | LASER (ACL 22) | COPNER$_{LiLT}$ (COLING 22) | **PPTSER$_{LiLT}$** (Ours) | COPNER$_{LMv3}$ (COLING 22) | **PPTSER$_{LMv3}$** (Ours) |
| FUNSD | 1-shot | 24.32 | 19.37 | 38.47 | 55.15 | 55.64 | 51.19 | **61.98** |
| | 3-shot | 34.94 | 31.21 | 44.88 | 68.66 | 69.17 | 75.84 | **76.86** |
| | 5-shot | 39.55 | 35.13 | 49.31 | 73.43 | 75.26 | 77.55 | **81.53** |
| | 7-shot | 41.41 | 37.31 | 52.56 | 73.35 | 75.71 | 78.53 | **81.60** |
| | Full Data | 67.42 | 64.58 | 69.23 | 87.74 | 89.07 | 91.26 | **92.01** |
| CORD-Lv1 | 1-shot | 74.29 | 68.61 | 66.80 | 86.97 | **90.50** | 86.98 | 90.02 |
| | 3-shot | 83.68 | 82.25 | 76.09 | 94.16 | 94.79 | 94.03 | **95.13** |
| | 5-shot | 87.11 | 86.08 | 82.23 | 94.86 | **96.21** | 95.74 | **96.21** |
| | 7-shot | 87.31 | 86.74 | 83.61 | 95.04 | 96.13 | 96.06 | **96.51** |
| | Full Data | 95.93 | 95.90 | 96.56 | 99.21 | 99.42 | **99.45** | **99.45** |
| CORD | 1-shot | 57.86 | 54.52 | - | 70.05 | **75.57** | 67.33 | 74.19 |
| | 3-shot | 71.68 | 71.32 | - | 81.27 | 83.83 | 80.07 | **85.27** |
| | 5-shot | 77.74 | 78.98 | - | 84.80 | 87.06 | 85.30 | **87.77** |
| | 7-shot | 78.63 | 78.63 | - | 85.76 | 87.73 | 86.87 | **88.48** |
| | Full Data | 93.50 | 94.16 | - | 95.74 | 96.04 | 95.79 | **96.39** |
| XFUND-zh | 1-shot | 26.38 | 23.29 | - | 48.76 | **67.64** | 54.26 | 56.71 |
| | 3-shot | 37.22 | 37.49 | - | 64.59 | 74.17 | 71.27 | **75.24** |
| | 5-shot | 43.54 | 44.36 | - | 69.03 | **79.40** | 76.37 | 79.26 |
| | 7-shot | 46.62 | 46.90 | - | 74.44 | **81.38** | 79.29 | 80.97 |
| | Full Data | 66.20 | 67.11 | - | 89.17 | 90.61 | 91.99 | **92.19** |

**Setup.** We selected the PPTSER models that performed better under different modality settings, denoted as **PPTSER$_{LiLT}$** and **PPTSER$_{LMv3}$**, and compared them with previous few-shot methods. For a comprehensive comparison, we re-implemented **LASER** (Wang & Shang, 2022) on our few-shot divisions. However, LASER can only handle the coarse-level typing for CORD (CORD-Lv1) and is limited to English language. Given that research on few-shot SER is rather limited, we selected two other few-shot NER methods for comparisons. Specifically, We chose **COPNER** (Huang et al., 2022a) and **EntLM** (Ma et al., 2022b) due to their similar in-label-space setting with ours. Considering COPNER can also be used as a pluggable method, we also implemented it with LayoutLMv3 and LiLT, denoted as **COPNER$_{LiLT}$** and **COPNER$_{LMv3}$**.

Table 3: Performances (%) of PPTSER on CORD benchmark using distinct strategies. *Pre.* and *Rec.* are the abbreviations for *Precision* and *Recall.* And *Avg. Diff.* refers to the average difference of the corresponding metrics compared to those in *entity typing & span detection*.

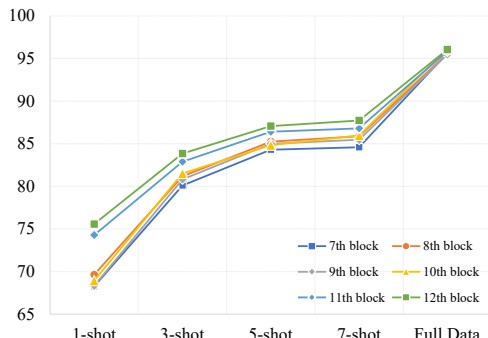

Figure 4: F1 score (%) of PPTSER on CORD benchmark with different settings when obtaining the attention weight from different blocks.

| | PPTSER | | | | | |
|---|---|---|---|---|---|---|
| | *entity typing & span detection* | | | *plain BIO prompt* | | |
| | Pre. | Rec. | F1 | Pre. | Rec. | F1 |
| 1-shot | 75.35 | 75.80 | 75.57 | 73.92 | 74.66 | 74.28 |
| 3-shot | 83.85 | 83.82 | 83.83 | 82.44 | 82.37 | 82.41 |
| 5-shot | 87.03 | 87.08 | 87.06 | 86.50 | 86.01 | 86.25 |
| 7-shot | 87.74 | 87.72 | 87.73 | 86.64 | 86.35 | 86.49 |
| Full Data | 96.06 | 96.03 | 96.04 | 95.58 | 95.00 | 95.29 |
| Avg. Diff. | - | - | - | -0.99 | -1.21 | -1.10 |

**Results.** The overall experimental results are presented in Table 2. The results clearly demonstrate that PPTSER outperforms existing few-shot NER and few-shot SER methods by a large margin. Surprisingly, COPNER shows some degree of pluggability when equipped with multi-modal pre-trained models, but PPTSER still outperforms it in all settings across each benchmark. For more experimental results, please refer to A.4.2.

In summary, our PPTSER surpasses existing few-shot NER and few-shot SER methods on various visually-rich documents, showcasing its effectiveness in handling few-shot SER challenge.

## 5 ANALYSIS

We have conducted extensive analyses of our PPTSER to ensure its effectiveness and rationality. For convenience, experiments are conducted on CORD benchmark using PPTSER building upon LiLT.

**Origin of Attention Weights.** To determine the source of superiority in PPTSER, we investigated whether it stems from our meticulous design or from the reduction of over-fitting achieved through the parameter reduction. We conducted experiments to obtain the attention weights from distinct blocks. In addition to the default *12th block*, we also extracted attention weights from the *7th ∼ 11th* block, and the experimental results are illustrated in Figure 4. It is shown that our design to obtain attention weight from the last block outperforms those to obtain it from other blocks, which has greatly assured the effectiveness of our design. For the detailed metrics, please refer to A.4.3.

**Effectiveness of Decoupling Strategies.** Table 3 shows the comparisons of utilizing different frameworks. In this context, *entity typing & span detection* refers to our design to decouple the SER task into entity typing and span detection, while *plain BIO prompt* refers to the direct usage of $\tilde{C} = \{e_0, \, beginning \, of \, e_i, \, inner \, of \, e_i\}$ as the prompt, focusing solely on entity typing. The result shows that decoupling SER as *entity typing & span detection* and processing them in parallel avoids interfering with the language modeling for the document tokens and gets better performance.

**Prompt Engineering.** We also evaluated our PPTSER with different types of prompts. Table 4 shows the results of testing PPTSER with various prompt types, including *unrelated words*, such as `apple` and `orange`, which are irrelevant to the SER task, *random embeddings* that uses randomly initialized tensors as the word embeddings for the prompt and our default *tag name*. We observe our default setting yields the highest score, indicating that the pre-trained model does learn the semantics of the prompt to some extent, and the semantics of tags could guide the SER task. This suggests that careful selection and crafting of prompts can significantly impact the performance of the model.

**Aggregation of Attention Weights.** Table 5 shows the performances of different strategies to aggregate the attention weights across distinct attention heads. In this context, *max* and *mean* refer to the maximum and average value across attention weighs of distinct heads, while *single head* refers to utilizing only a single head of the attention weights to generate the final probability. The results indicate that the *max* operation demonstrates the best performance, which aligns with our hypothesis that different attention heads focus on entities with different semantics, as shown in Figure 5.

Table 4: Performances (%) of PPTSER on CORD benchmark using different prompts. *Avg. Diff.* refers to the average difference of the corresponding metrics compared to those in *tag names*.

| | PPTSER | | | | | | | | |
|---|---|---|---|---|---|---|---|---|---|
| | *tag names* | | | *unrelated words* | | | *random embeddings* | | |
| | Pre. | Rec. | F1 | Pre. | Rec. | F1 | Pre. | Rec. | F1 |
| 1-shot | 75.35 | 75.80 | 75.57 | 70.58 | 71.03 | 70.80 | 71.32 | 71.54 | 71.43 |
| 3-shot | 83.85 | 83.82 | 83.83 | 82.83 | 82.54 | 82.69 | 83.24 | 82.90 | 83.07 |
| 5-shot | 87.03 | 87.08 | 87.06 | 85.84 | 85.72 | 85.78 | 86.47 | 86.11 | 86.29 |
| 7-shot | 87.74 | 87.72 | 87.73 | 86.44 | 86.40 | 86.42 | 87.09 | 86.81 | 86.95 |
| Full Data | 96.06 | 96.03 | 96.04 | 96.27 | 96.24 | 96.26 | 95.75 | 95.70 | 95.72 |
| Avg. Diff. | - | - | - | **-1.61** | **-1.70** | **-1.66** | **-1.23** | **-1.48** | **-1.35** |

Table 5: Performances (%) of PPTSER on CORD benchmark using different aggregation strategies for attention heads. *Avg. Diff.* refers to the average difference of the corresponding metrics compared to those in *max*.

| | PPTSER | | | | | | | | |
|---|---|---|---|---|---|---|---|---|---|
| | *max* | | | *mean* | | | *single head* | | |
| | Pre. | Rec. | F1 | Pre. | Rec. | F1 | Pre. | Rec. | F1 |
| 1-shot | 75.35 | 75.80 | 75.57 | 74.16 | 74.69 | 74.43 | 74.92 | 75.51 | 75.21 |
| 3-shot | 83.85 | 83.82 | 83.83 | 83.04 | 83.06 | 83.05 | 84.07 | 83.77 | 83.92 |
| 5-shot | 87.03 | 87.08 | 87.06 | 86.61 | 86.67 | 86.64 | 86.97 | 86.99 | 86.98 |
| 7-shot | 87.74 | 87.72 | 87.73 | 86.76 | 86.71 | 86.74 | 87.30 | 87.19 | 87.24 |
| Full Data | 96.06 | 96.03 | 96.04 | 96.23 | 96.18 | 96.21 | 96.08 | 96.03 | 96.06 |
| Avg. Diff. | - | - | - | **-0.64** | **-0.63** | **-0.63** | **-0.14** | **-0.19** | **-0.17** |

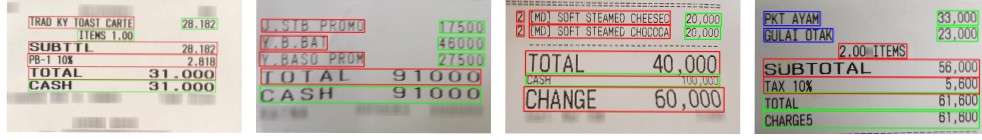

Figure 5: Illustration of different entity regions that were focused on by different attention heads on CORD benchmark. Different colored boxes represent the areas of focus of different attention heads.

**Parameter Efficiency.** The parameter comparisons of our PPTSER methods and traditional Fine-tuning are presented in Table 6. As the parameters might vary across different models and benchmarks, we offer a comprehensive breakdown of the results from the methods we have tested. The results illustrate that our PPTSER has fewer parameters in comparison to traditional fine-tuning methods. For a more detailed analysis, please refer to A.5.

Table 6: Parameters of our PPTSER and traditional Fine-tuning methods. The metric in **Bold** indicates the method with fewer parameters. *FT* refers to *Fine-tuning* method.

| Methodology | BROS (AAAI 22) | | LiLT (ACL 22) | | LayoutLMv2 (ACL 21) | | LayoutLMv3 (MM 22) | |
|---|---|---|---|---|---|---|---|---|
| | FT | **Ours** | FT | **Ours** | FT | **Ours** | FT | **Ours** |
| FUNSD | 108.91M | **103.59M** | 130.17M | **123.81M** | 200.29M | **194.38M** | 125.33M | **119.42M** |
| CORD | 108.95M | **103.59M** | 130.22M | **123.81M** | 200.33M | **194.38M** | 125.96M | **119.42M** |
| XFUND-zh | - | **103.59M** | 130.17M | **123.81M** | 200.29M | **194.38M** | 125.33M | **119.42M** |

## 6 CONCLUSION

In this paper, we introduced PPTSER, an innovative and efficient strategy for few-shot entity recognition in visually-rich documents using a plug-and-play, tag-guided approach. This was accomplished by redefining the SER task as a dual-function operation of entity typing and span detection, and utilizing prompts related to SER tags along with attention weight as the target probability distributions. Our results demonstrate that PPTSER is both effective and versatile in a variety of data settings, from few-shot to full data scenarios. In the future, we plan to further investigate the capabilities of PPTSER across a range of VIE tasks, including Entity Linking. In addition, we aim to explore the potential of PPTSER in other few-shot scenarios, particularly those outside of the In-Label-Space setting. It is our hope that our work will spark further research and advancements in the realm of Few-shot SER.

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

# A APPENDIX

Our Appendix is organized in the following manner:

- In A.1, we describe the algorithm to generate the few-shot divisions for our experiments.
- In A.2, we discuss some implementation details of our PPTSER and other few-shot methods with which we conducted comparisons.
- In A.3, we supply an example to better demonstrate how our PPTSER processes a visually-rich document.
- In A.4, we provide more detailed experimental results of our main experiments.
- In A.5, we offer a further analysis on the parameter count of our PPTSER and the traditional fine-tuning methods.

## A.1 Few-shot Divisions Generation

To cater to the real-world application scenarios, we have organized our few-shot divisions from the full training set as Algorithm 1. Our goal was to randomly select the minimum number of documents that satisfy the *N-way K-shot* requirement of *there are K documents containing entities of a specific type across N categories*.

It is worth noticing that in the context of few-shot SER on visually-rich documents, the few-shot setting of *N-way K-shot* signifies that *there are K documents containing entities of a specific type across N categories*, instead of *there are K entity spans for each of N entity types* for the setting of few-shot NER on plain texts.

---

**Algorithm 1:** Few-shot Divisions Generation

---

**Input:** Novel Dataset with the label space $\mathbb{C} = \{c_1, c_2, ..., c_N\}$, full training set $\mathcal{D}^{full}$
**Output:** N-way K-shot few-shot training set $\mathcal{D}^{train}$

1   $\mathcal{D}^{train} = \{\}$
2   Number of documents that contain entities of $c_i$ in $\mathcal{D}^{full}$: $Q = \{c_1 : 0, c_2 : 0, ..., c_N : 0\}$
3   Document set that contain entities of $c_i$ in $\mathcal{D}^{full}$: $R = \{c_1 : \{\}, c_2 : \{\}, ..., c_N : \{\}\}$
4   **for** $doc_i$ *in* $\mathcal{D}^{full}$ **do**
5      **for** $c_j$ *in* $\mathbb{C}$ **do**
6         **if** $doc_i$ *contain entities of* $c_j$ **then**
7            $Q[c_j] \mathrel{+}= 1$
8            $R[c_j].append(doc_i)$
9         **end**
10      **end**
11   **end**
12   $Q' = sorted(Q, \; key = lambda \; x : x[1])$
      $= \{c'_1 : n_1, c'_2 : n_2, ..., c'_N : n_N\} \; (n_1 \le n_2 \le ... \le n_N)$
13   Number of documents that contain entities of $c_i$ in $\mathcal{D}^{train}$: $S = \{c_1 : 0, c_2 : 0, ..., c_N : 0\}$
14   **for** $c'_i$ *in keys of* $Q'$ **do**
15      **for** $S[c'_i] < K$ **do**
16         **if** $R[c'_i]$ *is empty* **then**
17            break
18         **end**
19         Randomly select a document $doc_{candidate}$ from $R[c'_i]$
20         $R[c'_j].pop(doc_{candidate})$
21         **if** $doc_{candidate} \notin \mathcal{D}^{train}$ **then**
22            $\mathcal{D}^{train}.append(doc_{candidate})$
23            **for** $c_j \in \mathbb{C}$ **do**
24               **if** $doc_{candidate}$ *contain entities of* $c_j$ **then**
25                  $S[c_j] \mathrel{+}= 1$
26               **end**
27            **end**
28         **else**
29            continue
30         **end**
31      **end**
32   **end**

---

## A.2 IMPLEMENTATION DETAILS

### A.2.1 IMPLEMENTATION DETAILS OF PPTSER

We used one NVIDIA 3090 to fine-tune our model with AdamW optimizer. The learning rate is $5e-5$ with a warm up ratio of $0.1$, and we fine-tuned the model for $2000$ iterations with a batch size of $8$ by default. Besides the default augmentation strategies for images adopted in LayoutLMv2 and LayoutLMv3, we did not employ any additional augmentation strategies.

### A.2.2 MODIFICATION ON FEW-SHOT NER METHOD FOR VISUALLY-RICH DOCUMENTS

In Section 4.3, we mentioned that we adapted two methods originally used for few-shot NER on plain text for few-shot SER on visually-rich documents. We will briefly introduce these modifications.

**COPNER (Huang et al., 2022a).** The COPNER method employs contrastive learning, feeding both entity label semantics and sentences into a plain text pre-trained language model. This approach uses the hidden state output of the pre-trained model to calculate a contrastive loss between sentence tokens and label semantics, then determining the entity type of tokens. However, the original COPNER could only determine if a token belonged to an entity category, without recognizing boundaries between entities. Therefore, we also improved it with the *entity typing and span detection* framework introduced in our paper. That is, while determining the entity type of tokens, we also input the tokens $beginning$ and $inner$ into the pre-trained model. The model's output hidden state is then used to calculate a contrastive loss between sentence tokens and these $beginning$ and $inner$ tokens.

Besides, we retained this core process but replaced the original language model pre-trained on pure text with a multi-modal pre-trained model. Experiments show that our use of multi-head cross-attention methods is more suitable for SER tasks on visually-rich documents, especially in Chinese contexts.

**EntLM (Ma et al., 2022b).** EntLM treats NER as a task of Language Modeling. For testing a few-shot NER dataset on plain text, it first selects a related word for each entity type. Then, using the pre-trained Masked Language Modeling head of BERT, it predicts the probability distribution of each sentence token over these related words, thereby determining the probability distribution of tokens across different entity types.

The selection of related words relies on the distant data obtained from BOND (Liang et al., 2020), which uses BERT and the corpora from Wikipedia to create rough annotations for the NER test set. However, in the realm of visually-rich documents, such distant data is not provided by BOND, and due to the relative abstract expression of SER tags from natural language expressions and the difference between structured documents and natural language expressions, it's not feasible to obtain rough annotations using corpora from Wikipedia with BERT. Therefore, we directly use the ground truth annotations from the SER test set as distant data to find related words associated with entity types in the SER dataset. Although the experimental results on EntLM might be artificially high due to some exposure to the entity distribution in the test set, our proposed method significantly outperforms others that only accept text modality inputs, such as EntLM.

In summary, methods for few-shot NER on plain text may not necessarily transition well to the task of few-shot SER on visually-rich documents. The notable performance of our proposed method in few-shot SER on visually-rich documents further highlights the innovation and contribution of our research.

A.3   EXAMPLE WHEN APPLY PPTSER

To more intuitively demonstrate our method, we provide an example from FUNSD dataset when apply our PPTSER method.

The FUNSD dataset includes three meaningful entity types: *header*, *question*, and *answer*, with all other uninteresting entities categorized as *other*. Hence, the entity type set is $\mathbb{E} = \{other, header, question, answer\}$. The label spaces for entity typing and span detection are $\mathbb{C}^{ent.} = \{other, header, question, answer\}$ and $\mathbb{C}^{det.} = \{beginning, inner\}$, respectively. We directly use the label's names as tag-related prompts, with the prompts for entity typing and span detection being $\tilde{\mathbb{C}}^{ent.} = "other\ header\ question\ answer"$ and $\tilde{\mathbb{C}}^{det.} = "beginning\ inner"$. These prompts are then concatenated to form the full tag-related prompt $\tilde{\mathbb{C}} = "other\ header\ question\ answer\ beginning\ inner"$.

Consider an example from the FUNSD dataset shown in Figure 6, with the document content *"... CASE TYPE: Asbestos ... 82504862"*, where *"..."* indicates omitted parts. Here, *"CASE TYPE:"* belongs to the entity type of *question*, *"Asbestos"* to the entity type of *answer*, and *"82504862"* to *other*. Assuming the tokenizer splits the document into *"CASE"*, *"TYPE:"*, *"Asbestos"*, and *"82504862"*, their labels for entity typing and span detection would be $\boldsymbol{y}^{ent.} = [question, question, answer, other]$ and $\boldsymbol{y}^{det.} = [beginning, inner, beginning, -1]$.

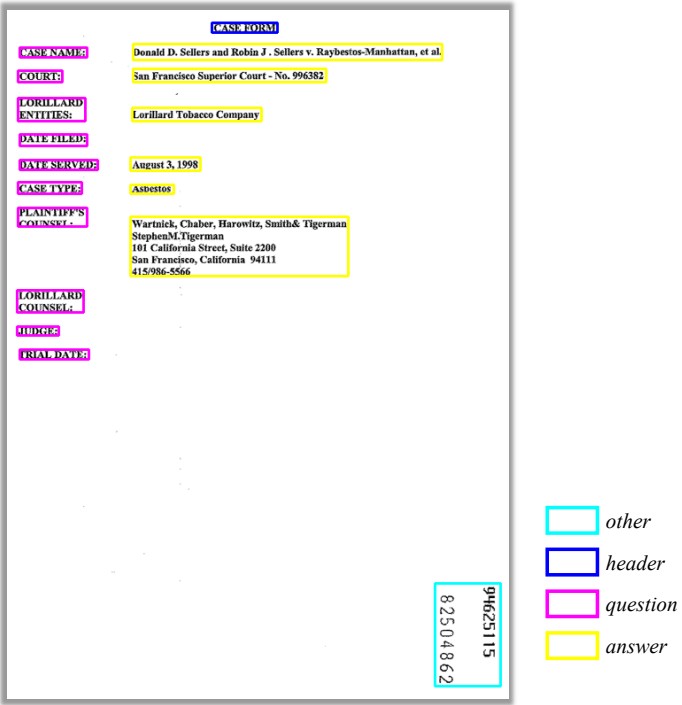

Figure 6: Illustration of a sample from FUNSD dataset. Different colored boxes represent the entities of distinct kinds. Zoom in for better view.

Subsequently, *"CASE"*, *"TYPE:"*, *"Asbestos"*, and *"82504862"* as document tokens are concatenated with the full tag-related prompt, and form the encoder input $\boldsymbol{x}' = "CASE\ TYPE : Asbestos\ 82504862\ other\ header\ question\ answer\ beginning\ inner"$. Then, $\boldsymbol{x}'$ is input into the multi-modal pre-trained model to obtain the multi-head attention weight and the aggregated attention weight from the last layer, as shown in Figure 7.

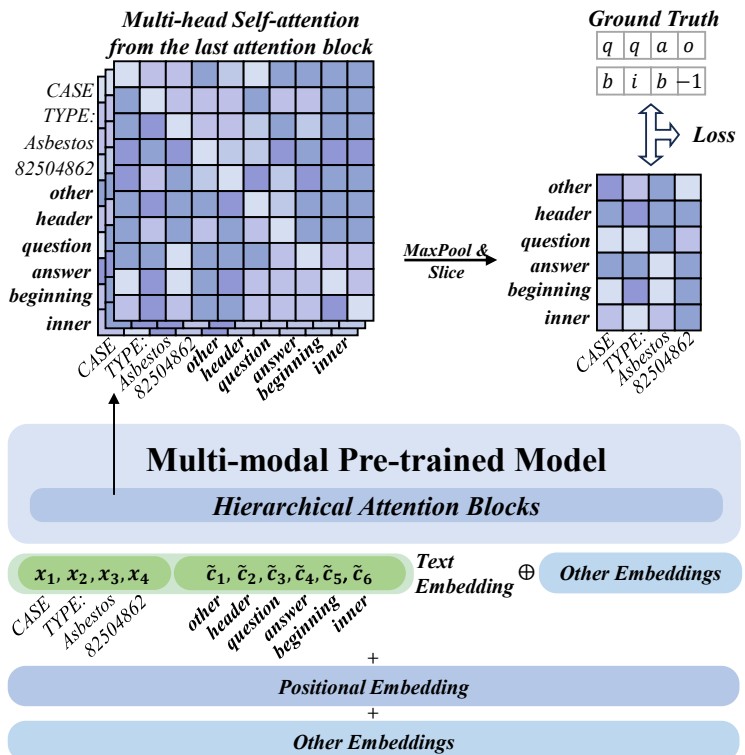

Figure 7: Example of a FUNSD sample running on PPTSER. In the image, bolded words in **Text Embeddings** indicate they are part of the prompt. In the Multi-head Self-attention section, the brightness of the color represents the magnitude of the value. In the Ground Truth, $q$, $a$, $b$, $i$ and $o$ respectively stand for the *question*, *answer*, *beginning*, *inner* and *other* categories.

During training stage, as shown in Figure 8(a), the aggregated attention weight is split into attention weights between "*other header question answer*" and document tokens, as well as "*beginning inner*" and document tokens, which are then used to calculate the losses for entity typing and span detection, respectively, culminating in a combined total loss.

During testing stage, as shown in Figure 8(b), we select the document tokens with the highest probability for "*other header question answer*" and "*beginning inner*" as $\hat{y}^{ent.}$ and $\hat{y}^{det.}$, then combine them to get the predictions under BIO tagging scheme $\hat{y}$ following the procedure in Section 3.2.2. And the entity spans are finally analysed from $\hat{y}$.

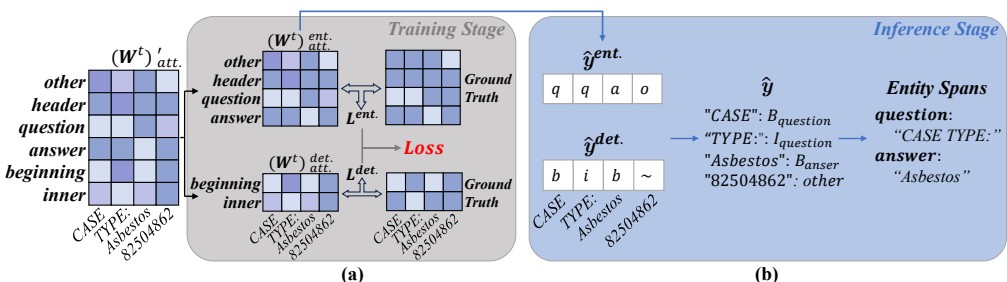

Figure 8: Example of a FUNSD sample running on PPTSER at *Training stage* and *Inference stage*.

## A.4 DETAILED EXPERIMENTAL RESULTS

This section presents additional performance metrics obtained from the main experiments.

### A.4.1 DETAILED RESULTS OF COMPARISONS WITH EXISTING FINE-TUNING METHODS

Table 7(a) and Table 7(b) present the precision and recall of PPTSER compared to the traditional fine-tuning method. The results demonstrate that PPTSER consistently outperforms the traditional fine-tuning method in most cases, leading to improved overall performance in terms of F1 scores.

Table 7: Precision and Recall of PPTSER and Traditional Fine-tuning methods.

(a) Precision (%) of our PPTSER and Traditional Fine-tuning methods. Precision in **Bold** is better between PPTSER and Fine-tuning. *FT* refers to *Fine-tuning* methods.

| Modality | | Text + Layout | | | | Text + Layout + Image | | | |
|---|---|---|---|---|---|---|---|---|---|
| Methodology | | BROS (AAAI 22) | | LiLT (ACL 22) | | LayoutLMv2 (ACL 21) | | LayoutLMv3 (MM 22) | |
| | | FT | **Ours** | FT | **Ours** | FT | **Ours** | FT | **Ours** |
| FUNSD | 1-shot | 49.17 | **52.50** | **51.07** | 50.91 | 44.15 | **46.91** | 42.67 | **56.27** |
| | 3-shot | 63.07 | **64.31** | 65.65 | **67.24** | **60.67** | 60.65 | 72.66 | **75.18** |
| | 5-shot | 66.31 | **68.10** | 71.11 | **72.95** | 63.45 | **64.36** | 77.29 | **79.53** |
| | 7-shot | 69.24 | **71.06** | 72.49 | **74.14** | 65.94 | **66.79** | 79.22 | **81.31** |
| | Full Data | 83.42 | **83.67** | 88.62 | **88.89** | 83.54 | **83.59** | 91.41 | **91.96** |
| CORD | 1-shot | 64.92 | **68.04** | 69.31 | **75.35** | 68.08 | **69.73** | 70.03 | **74.02** |
| | 3-shot | 78.75 | **79.26** | 81.63 | **83.85** | 79.92 | **81.35** | 81.97 | **85.09** |
| | 5-shot | 83.86 | **84.23** | 85.62 | **87.03** | 83.84 | **84.42** | 85.76 | **87.71** |
| | 7-shot | 83.56 | **83.76** | 85.39 | **87.74** | 84.40 | **85.09** | 86.92 | **88.37** |
| | Full Data | 95.72 | **95.88** | 95.82 | **96.06** | 94.95 | **95.64** | 96.34 | **96.39** |
| XFUND-zh | 1-shot | - | - | 59.03 | **64.90** | 59.17 | **67.04** | 46.64 | **54.59** |
| | 3-shot | - | - | 71.33 | **71.54** | 73.17 | **75.21** | 62.73 | **72.36** |
| | 5-shot | - | - | 75.09 | **77.07** | 79.21 | **79.96** | 69.39 | **76.57** |
| | 7-shot | - | - | 77.34 | **77.70** | 79.68 | **80.22** | 72.03 | **77.95** |
| | Full Data | - | - | 87.92 | **88.17** | 88.60 | **88.95** | 89.09 | **91.04** |

(b) Recall (%) of our PPTSER and Traditional Fine-tuning methods. Recall in **Bold** is better between PPTSER and Fine-tuning. *FT* refers to *Fine-tuning* methods.

| Modality | | Text + Layout | | | | Text + Layout + Image | | | |
|---|---|---|---|---|---|---|---|---|---|
| Methodology | | BROS (AAAI 22) | | LiLT (ACL 22) | | LayoutLMv2 (ACL 21) | | LayoutLMv3 (MM 22) | |
| | | FT | **Ours** | FT | **Ours** | FT | **Ours** | FT | **Ours** |
| FUNSD | 1-shot | 49.75 | **57.89** | 54.39 | **61.77** | 53.64 | **59.15** | 53.14 | **70.07** |
| | 3-shot | 65.82 | **71.50** | 69.81 | **71.35** | 62.82 | **67.01** | 76.94 | **78.64** |
| | 5-shot | 69.32 | **73.43** | 75.61 | **77.76** | 68.50 | **70.97** | 81.92 | **83.65** |
| | 7-shot | 67.54 | **72.94** | 74.33 | **77.38** | 67.23 | **71.04** | 80.51 | **81.91** |
| | Full Data | **84.26** | 84.15 | **89.30** | 89.26 | 83.51 | **83.85** | 90.91 | **92.05** |
| CORD | 1-shot | 67.70 | **68.95** | 70.79 | **75.80** | **71.23** | 70.22 | 70.67 | **74.35** |
| | 3-shot | 79.30 | **79.96** | 81.65 | **83.82** | 81.36 | **81.98** | 82.14 | **85.45** |
| | 5-shot | 84.21 | **84.52** | 85.43 | **87.08** | **84.81** | 84.63 | 85.91 | **87.84** |
| | 7-shot | 83.79 | **84.43** | 85.32 | **87.72** | 85.12 | **85.54** | 86.97 | **88.59** |
| | Full Data | **95.72** | 95.61 | 95.78 | **96.03** | 95.45 | **95.62** | 96.34 | **96.40** |
| XFUND-zh | 1-shot | - | - | 61.54 | **70.68** | 61.46 | **69.68** | **61.21** | 59.24 |
| | 3-shot | - | - | 74.22 | **77.22** | 75.88 | **79.46** | 77.24 | **78.58** |
| | 5-shot | - | - | 79.99 | **82.02** | 83.88 | **85.02** | **82.27** | 82.23 |
| | 7-shot | - | - | 83.97 | **85.44** | 85.03 | **87.43** | **84.76** | 84.30 |
| | Full Data | - | - | 93.18 | **93.20** | 91.96 | **92.72** | **94.27** | 93.37 |

### A.4.2 DETAILED RESULTS OF COMPARISONS WITH EXISTING FEW-SHOT METHODS

We also present a detailed comparison of PPTSER with other few-shot methods, including precision and recall metrics in Table 8(a) and Table 8(b). Similar to the F1 score, models enhanced with PPTSER usually demonstrate superior performance compared to both few-shot NER and few-shot SER methods.

Table 8: Precision and Recall of PPTSER and other Few-shot methods.

(a) Precision (%) of PPTSER and other Few-shot methods. Precision in **Bold** is the best, and that with underline is the second best.

| Modality | | Text | | Text + Layout | | | Text + Layout + Image | |
|---|---|---|---|---|---|---|---|---|
| Methodology | | EntLM (NAACL 22) | COPNER (COLING 22) | LASER (ACL 22) | COPNER$_{LiLT}$ (COLING 22) | **PPTSER$_{LiLT}$** (Ours) | COPNER$_{LMv3}$ (COLING 22) | **PPTSER$_{LMv3}$** (Ours) |
| FUNSD | 1-shot | 22.85 | 18.67 | 36.61 | 53.79 | 50.91 | 49.01 | **56.27** |
| | 3-shot | 33.39 | 30.97 | 46.71 | 67.26 | 67.24 | 73.54 | **75.18** |
| | 5-shot | 37.23 | 32.57 | 46.80 | 71.40 | 72.95 | 75.40 | **79.53** |
| | 7-shot | 40.29 | 35.43 | 51.17 | 72.81 | 74.14 | 78.63 | **81.31** |
| | Full Data | 67.53 | 63.39 | 69.08 | 87.45 | 88.89 | 91.45 | **91.96** |
| CORD-Lv1 | 1-shot | 73.23 | 67.42 | 65.56 | 86.93 | **90.62** | 86.80 | 90.05 |
| | 3-shot | 82.85 | 81.37 | 75.43 | 94.39 | 94.97 | 93.99 | **95.14** |
| | 5-shot | 86.87 | 85.87 | 82.07 | 94.94 | **96.38** | 95.65 | 96.20 |
| | 7-shot | 86.65 | 86.54 | 83.54 | 95.12 | 96.25 | 96.05 | **96.53** |
| | Full Data | 95.93 | 95.83 | 96.50 | 99.23 | 99.43 | **99.45** | **99.45** |
| CORD | 1-shot | 57.45 | 51.22 | - | 70.27 | **75.35** | 67.14 | 74.02 |
| | 3-shot | 71.42 | 67.26 | - | 81.48 | 83.85 | 79.89 | **85.09** |
| | 5-shot | 77.70 | 74.59 | - | 84.92 | 87.03 | 85.13 | **87.71** |
| | 7-shot | 78.51 | 75.69 | - | 85.86 | 87.74 | 86.83 | **88.37** |
| | Full Data | 93.56 | 92.33 | - | 95.75 | 96.06 | 95.79 | **96.39** |
| XFUND-zh | 1-shot | 27.02 | 23.98 | - | 49.49 | **64.90** | 53.37 | 54.59 |
| | 3-shot | 35.94 | 35.72 | - | 64.69 | 71.54 | 69.98 | **72.36** |
| | 5-shot | 43.18 | 42.93 | - | 68.21 | **77.07** | 73.48 | 76.57 |
| | 7-shot | 45.13 | 44.66 | - | 72.61 | 77.70 | 76.42 | **77.95** |
| | Full Data | 64.75 | 65.59 | - | 87.23 | 88.17 | **91.10** | 91.04 |

(b) Recall (%) of PPTSER and other Few-shot methods. Recall in **Bold** is the best, and that with underline is the second best.

| Modality | | Text | | Text + Layout | | | Text + Layout + Image | |
|---|---|---|---|---|---|---|---|---|
| Methodology | | EntLM (NAACL 22) | COPNER (COLING 22) | LASER (ACL 22) | COPNER$_{LiLT}$ (COLING 22) | **PPTSER$_{LiLT}$** (Ours) | COPNER$_{LMv3}$ (COLING 22) | **PPTSER$_{LMv3}$** (Ours) |
| FUNSD | 1-shot | 28.01 | 21.53 | 41.05 | 56.72 | 61.77 | 54.43 | **70.07** |
| | 3-shot | 37.56 | 32.12 | 46.55 | 70.17 | 71.35 | 78.32 | **78.64** |
| | 5-shot | 42.42 | 38.28 | 52.14 | 75.59 | 77.76 | 79.83 | **83.65** |
| | 7-shot | 42.79 | 39.57 | 54.08 | 73.92 | 77.38 | 78.45 | **81.91** |
| | Full Data | 67.31 | 65.81 | 69.41 | 88.04 | 89.26 | 91.06 | **92.05** |
| CORD-Lv1 | 1-shot | 75.38 | 69.90 | 68.12 | 87.01 | **90.38** | 87.16 | 89.99 |
| | 3-shot | 84.54 | 83.17 | 76.77 | 93.95 | 94.61 | 94.06 | **95.12** |
| | 5-shot | 87.35 | 86.30 | 82.42 | 94.78 | **96.04** | 95.84 | 96.23 |
| | 7-shot | 87.99 | 86.97 | 83.71 | 94.96 | 96.00 | 96.07 | **96.50** |
| | Full Data | 95.94 | 95.97 | 96.62 | 99.19 | 99.42 | **99.45** | **99.45** |
| CORD | 1-shot | 58.29 | 53.79 | - | 69.83 | **75.80** | 67.51 | 74.35 |
| | 3-shot | 71.95 | 68.35 | - | 81.08 | 83.82 | 80.24 | **85.45** |
| | 5-shot | 77.78 | 74.69 | - | 84.68 | 87.08 | 85.47 | **87.84** |
| | 7-shot | 78.76 | 76.51 | - | 85.66 | 87.72 | 86.91 | **88.59** |
| | Full Data | 93.45 | 92.68 | - | 95.72 | 96.03 | 95.79 | **96.40** |
| XFUND-zh | 1-shot | 26.35 | 23.26 | - | 48.26 | **70.68** | 55.67 | 59.24 |
| | 3-shot | 39.62 | 40.48 | - | 64.88 | 77.22 | 72.89 | **78.58** |
| | 5-shot | 44.03 | 46.14 | - | 70.00 | 82.02 | 79.92 | **82.23** |
| | 7-shot | 48.46 | 49.58 | - | 76.58 | **85.44** | 82.47 | 84.30 |
| | Full Data | 67.72 | 68.71 | - | 91.20 | 93.20 | 92.91 | **93.37** |

A.4.3   DETAILED RESULTS OF ATTENTION WEIGHTS OBTAINED FROM DIFFERENT BLOCKS

We also provide the numerical metrics of distinct settings to obtain the attention weight from different blocks. Table 9 illustrates the experimental results, indicating that obtaining attention weights from the last block yields the best performance of F1 score, precision and recall. Although the reduction of parameters alleviates over-fitting to some extent, since some shallower blocks outperform certain deeper ones in 1-shot scenario, our default setting to obtain the attention weight from the last block significantly outperforms the alternative settings of obtaining the attention weight from shallower blocks. This finding strongly reinforces the effectiveness of our design.

Table 9: Performances of PPTSER on CORD benchmark when obtaining the attention weight from different blocks.

(a) F1 score (%) of PPTSER on CORD benchmark when obtaining the attention weight from different blocks. *Avg. Diff.* refers to the average difference of F1 score compared to that in *12th block*.

| | PPTSER | | | | | |
|---|---|---|---|---|---|---|
| | *7th block* | *8th block* | *9th block* | *10th block* | *11th block* | *12th block* |
| 1-shot | 68.27 | 69.63 | 68.34 | 68.93 | 74.27 | 75.57 |
| 3-shot | 80.09 | 81.16 | 80.81 | 81.51 | 82.89 | 83.83 |
| 5-shot | 84.29 | 85.25 | 85.00 | 84.84 | 86.41 | 87.06 |
| 7-shot | 84.58 | 85.86 | 85.48 | 85.97 | 86.79 | 87.73 |
| Full Data | 95.63 | 95.59 | 95.48 | 96.12 | 95.86 | 96.04 |
| Avg. Diff. | **-3.47** | **-2.55** | **-3.03** | **-2.57** | **-0.80** | - |

(b) Precision (%) of PPTSER on CORD benchmark when obtaining the attention weight from different blocks. *Avg. Diff.* refers to the average difference of Precision compared to that in *12th block*.

| | PPTSER | | | | | |
|---|---|---|---|---|---|---|
| | *7th block* | *8th block* | *9th block* | *10th block* | *11th block* | *12th block* |
| 1-shot | 67.36 | 69.02 | 67.86 | 68.53 | 73.91 | 75.35 |
| 3-shot | 79.76 | 81.19 | 80.78 | 81.46 | 82.76 | 83.85 |
| 5-shot | 80.98 | 85.38 | 85.01 | 84.78 | 86.22 | 87.03 |
| 7-shot | 84.41 | 85.92 | 85.52 | 85.97 | 86.68 | 87.74 |
| Full Data | 95.70 | 95.64 | 95.55 | 96.16 | 95.87 | 96.06 |
| Avg. Diff. | **-4.36** | **-2.58** | **-3.06** | **-2.63** | **-0.92** | - |

(c) Recall (%) of PPTSER on CORD benchmark when obtaining the attention weight from different blocks. *Avg. Diff.* refers to the average difference of Recall compared to that in *12th block*.

| | PPTSER | | | | | |
|---|---|---|---|---|---|---|
| | *7th block* | *8th block* | *9th block* | *10th block* | *11th block* | *12th block* |
| 1-shot | 69.21 | 70.25 | 68.83 | 69.34 | 74.64 | 75.80 |
| 3-shot | 80.43 | 81.14 | 80.84 | 81.56 | 83.03 | 83.82 |
| 5-shot | 80.41 | 85.13 | 84.99 | 84.89 | 86.60 | 87.08 |
| 7-shot | 84.75 | 85.80 | 85.43 | 85.97 | 86.90 | 87.72 |
| Full Data | 95.57 | 95.54 | 95.41 | 96.09 | 95.85 | 96.03 |
| Avg. Diff. | **-4.02** | **-2.52** | **-2.99** | **-2.52** | **-0.69** | - |

### A.5 PARAMETER ANALYSIS OF PPTSER OVER TRADITIONAL FINE-TUNING

We provide a further analysis of the parameter counts in this section. As shown in Table 6, our PPTSER maintains consistent parameters across different benchmarks with the same pre-trained model. This is attributed to the fact that PPTSER does not necessitate an additional classifier layer, unlike the traditional fine-tuning method. Consequently, the parameter variance arises when employing the traditional fine-tuning method with the same pre-trained models on different benchmarks, owing to variations in the number of entity types present in those benchmarks. Furthermore, as PPTSER omits the value transform layer and the feed-forward layer in the final attention block, we achieve a reduction in the parameter count of the pre-trained model it is based on. Additionally, the extent of parameter reduction varies among different pre-trained models due to disparities in their architectural designs, resulting in slice differences in the eliminations of the modules.

