# OpenReview forum: "PPTSER: A Plug-and-Play Tag-guided Method for Few-shot Semantic Entity Recognition on Visually-rich Documents"
_ICLR.cc/2024/Conference — Submitted to ICLR 2024_

### Official Review · Reviewer_1jv5 · 2023-10-28

**Soundness:** 2 fair
**Presentation:** 2 fair
**Contribution:** 1 poor
**Rating:** 3
**Confidence:** 4

**Summary:**

This paper presents a plug-like label-guided method for few-shot entity recognition in visually rich documents. The approach leverages the semantics of tags to guide the SER task, resulting in good performance even with limited data. The method surpasses fine-tuned baselines and existing few-shot methods on various few-shot benchmarks.

**Strengths:**

1.	The proposed method demonstrates effectiveness in few-shot VRD entity recognition tasks.

2.	The approach is intriguing as it utilizes the semantics of tags to guide the SER task.

3.	The experimental results showcase that the proposed method outperforms fine-tuning baselines and other existing few-shot SER methods, particularly on the FUNSD and XFUND-zh benchmarks.

**Weaknesses:**

1.	The novelty is needed to be justified just as the motivation is not clear given the proposed architecture utilizing cross-attention has been widely discussed in NER for pure text.

2.	Does the issue of label leakage exist in the current training and testing scheme? Whether the prompts are also used again in testing?

3.	The figures in the paper exhibit a significant amount of overlap and difficult to understand. The presentation way needs to improve.

**Questions:**

1.	While the author claims to have addressed the challenge of the “In-Label-Space setting for few-shot SER,” this challenge is not adequately introduced in this paper. Furthermore, it is crucial to acknowledge that the In-Label-Space setting may deviate from the real-world challenges encountered in few-shot/zero-shot/meta-learning scenarios. In most cases, our primary interest in this field lies in enabling machines to learn novel entity types rather than knowing “B-XX” to infer “I-XX”.

2.	Since Figure 2 appears to be less informative on its own, a better approach would indeed be to combine Figure 1(b) and Figure 2. This combination would provide a clearer representation of the reputation within the context of these two figures.

3.	Regarding LayoutLMs and aligning multimodal inputs, it is important to consider how spatial/visual embeddings are handled for tag-related prompt tokens.

4.	In Figure 2, there are two parts labeled as “other embeddings.” It would be helpful to understand the distinction between these two parts.

5.	The datasets used in this study, such as FUNSD and XFUND-zh, are indeed relatively small and contain only a few entity types (e.g., only 3 in FUNSD). This limitation makes it challenging to fully assess the effectiveness of the In-Label-Space setting for few-shot SER on these specific datasets. Using CORD is suitable, but not enough.

6.	The paper mentions that “words related to SER tags are used as a prompt,” but it is not adequately explained what the tag-related prompt actually contains or how it is constructed. It would be beneficial for the author to provide some examples or utilize a running example prompt to illustrate the training process more clearly. This would help readers better understand what occurs during training and how the tag-related prompt influences the model’s performance.

7.	Table 1 shows marginal improvements when using the full data but significant improvement when using only a few instances. However, for CORD, the improvement is not as significant. The reason behind this discrepancy is unclear and requires further investigation.

---

> ### Author Response · Authors · 2023-11-20
> **Response to Reviewer 1jv5 (1/4)**
>
> Thank you for your patient review. Our following responses will address your concerns.
> ## Weakness
>
> - **Weakness #1: Novelty Concerns**
>    - We appreciate the observation regarding the application of cross-attention in NER for pure text. However, **our approach to implementing cross-attention for few-shot SER is unique**; it leverages the intrinsic self-attention mechanism within a pre-trained Transformer model, as detailed in `Section 3.2.2`. Unlike other methods that employ an additional cross-attention module, our design relies solely on the core self-attention operation of the transformer architecture, distinguishing it from conventional methods. Different from many other approaches that use the hidden state derived from cross-attention for classification-like operations, **we directly use the attention weights between document tokens and tag-related prompts as the probability distribution of tokens belonging to distinct SER tags**. This approach eliminates the conventional operation of multiplying attention weights with the value layer and then feeding the output to a feed-forward layer, saving parameters and computational time, as described in `contributions` in `Section 1` and the subsection of `Parameter Efficiency` in `Section 5`.
>    - Furthermore, **we have also innovatively utilized the multi-head attention design of the attention mechanism**. By selecting the maximum value from different attention heads, we ensure that each head focuses on different aspects. This also allows for a mutual backup function between different attention heads, which is particularly suitable for the entity-rich scenario on visually-rich documents. The effectiveness of this design is validated in `Aggregation of Attention Weights` of `Section 5`.
>    - Importantly, **methods used for NER on plain text may not necessarily be applicable to visually-rich documents**. As shown in `Table 2`, we compared some modified NER methods for pure text with our method in the context of SER on visually-rich documents. The results clearly show that our method outperforms the modified pure text methods, underlining a clear distinction between SER on visually-rich documents and NER on pure text and reinforcing the significance of our research in this field.
>    - Furthermore, multi-modal pre-trained models, typically pre-trained on structured documents, differ from the corpora used to pre-train BERT, which is closer to natural language expressions. Hence, **exploring whether these multi-modal pre-trained models can understand the semantics of SER tags, which is closer to natural language expressions, and use them to guide the SER task is a valuable question.** Our approach has proven effective across various multi-modal pre-trained models.
>    - Also, **our method could effectively type the entities and identify the boundaries between them**. We split SER on visually-rich documents into two distinct tasks: entity typing and span detection. Innovatively, we explored the feasibility of using natural language descriptions as prompts (*beginning* and *inner*) for span detection, allowing us to simultaneously handle both tasks within a single framework.
>
> - **Weakness #2: Questions on label leakages**
>
>    We appreciate your concern, but it seems there may be a slight misunderstanding regarding our experimental setup. As mentioned in `Section 3.1` of our paper, we employ an **In-Label-Space** setting where the label space for both the training set and test set is identical. Hence, they use the same tag-related prompt. However, this does not imply a label leakage issue in our study. Our research aims to improve the model's ability to efficiently extract information from novel types of visually-rich documents, such as invoices, receipts, etc., with minimal data annotation. It is not focused on extracting new entity types from existing visually-rich documents. The challenge is to design the model to handle a novel document scenario with limited labelled data, not to introduce novel entity types. Therefore, the use of the same prompts in testing does not lead to label leakage but rather aligns with our experimental design and objectives.
>
> - **Weakness #3 and Question #2: Questions on the representation of figures**
>
>    We are grateful for your suggestion. `Figure 1` was designed to provide an intuitive comparison between our PPTSER method and the conventional fine-tuning approach in terms of parameters and structure. However, we recognize that this could lead to a potential content overlap between `Figure 1(b)` and `Figure 2`.
>
>    In response to your comment, we have made preliminary modifications to the PDF version of our paper and are still actively seeking a more effective way to present this information. Rest assured, we will revise and update our paper accordingly in the final version to ensure clarity and ease of understanding.

---

> > ### Author Response · Authors · 2023-11-20
> > **Response to Reviewer 1jv5 (2/4)**
> >
> > ## Question
> >
> > - **Question #1: In-Label-Space setting concerns**
> >
> >    - Thank you for your comment. We understand your perspective, **but we respectfully disagree that the In-Label-Space setting deviates from real-world challenges**. The In-Label-Space configuration is particularly relevant in the context of few-shot SER on visually-rich documents. It is commonly used in various other few-shot SER methods. As we've described in `Section 3.1` and our response to `Weakness 2`, In-Label-Space few-shot setting, where the label space for both the training and test sets are identical, aiming to achieve good results on the test set with as few training set samples as possible. In visually-rich documents, when we encounter a new document scenario, our entity categories are generally predefined, and it's uncommon to introduce new categories. Our goal is to annotate as few documents as possible while still enabling the model to perform effectively. As discussed in the subsection `Few-shot SER on Visually-rich Documents` in `Section 2`, many other few-shot SER methods on visually-rich documents adopt a similar In-Label-Space setting, including works [Wang and Shang](https://aclanthology.org/2022.findings-acl.329/), [Yao et al.](https://arxiv.org/abs/2109.13967), and [Zanfir and Sminchisescu](https://dl.acm.org/doi/abs/10.1145/3394171.3413511). Furthermore, some multi-modal pre-trained models also use the In-Label-Space approach to demonstrate their few-shot capabilities and address the issue of few-shot SER, such as [Hong et al.](https://ojs.aaai.org/index.php/AAAI/article/view/21322) Even in the field of few-shot NER for pure text, there are similar studies [Huang et al.](https://aclanthology.org/2022.findings-acl.329/) and [Ma et al.](https://aclanthology.org/2022.naacl-main.420.pdf) adopting the same In-Label-Space setting. Therefore, we consider our In-Label-Space setting to be of great importance. We acknowledge that enabling the model to learn novel entity types holds potential application value, and this will be a direction for our future efforts.
> >    - Furthermore, **we contend that accurately identifying the boundaries between entities is of paramount importance in the SER task on visually-rich documents.** In the context of NER on plain text, the demarcation of boundaries between entities has not been sufficiently emphasized. However, in SER tasks on visually-rich documents, distinguishing the boundaries of entities accurately is crucial. Due to layout and other factors, the sequence of text on visually-rich documents input to the model may not align with the natural reading order. This can lead to situations where two entities of the same category, if not properly delineated, could be mistakenly merged into a single incorrect entity. For instance, consider two entities of the same type, '3.14' and '2.72'. Suppose their input sequence to the model is '3 . 1 4 2 . 7 2'. When employing the IO tagging scheme from NER on plain text, which does not differentiate boundaries, the predictions would be *'I I I I I I I I'* (Where *'I'* stands for the inner of an entity), leading to the incorrect single entity '3.142.72'. Our method, however, utilizes span detection to fit the BIO tagging scheme, which is able to yield the predictions *'B I I I B I I I*' (Where *'B'* stands for the beginning of an entity), successfully identifying '3.14' and '2.72' as separate entities.
> >
> > - **Question #3: Settings on spatial/visual embeddings**
> >
> >    Thank you for your thoughtful question. You're correct in observing that the tag-related prompts we use are not actually present in the tokens from the visually-rich documents. As a result, we have currently set their spatial and visual embeddings to zero.
> >
> >    We have conducted experiments where we treated the spatial embeddings of the prompts as learnable tensors. However, our preliminary experiments did not show significant effects, which led us to not focus our main efforts on this approach subsequently.
> >
> >    That being said, the impact of the settings for spatial/visual embeddings on our experimental results is indeed an intriguing issue. We acknowledge its importance and plan to further explore the settings of these two types of embeddings in our future research.

---

> > > ### Author Response · Authors · 2023-11-20
> > > **Response to Reviewer 1jv5 (3/4)**
> > >
> > > ## Question
> > > - **Question #4: Confusions on *Other embeddings***
> > >
> > >    The purpose of setting the two *Other Embeddings* is to comprehensively cover the multi-modal pre-trained models we use, such as LayoutLMv3 and LiLT. For instance, in the case of LayoutLMv3, image patches and document tokens are concatenated and then fed into the multi-modal pre-training model simultaneously; at the same time, an embedding is required to indicate whether the current position belongs to a text token embedding or an image patch embedding. Therefore, in this context, the *Other Embeddings* in the top right corner represent *Image Patch Embeddings*, and those located below the Positional Embedding are *Type Embeddings* that indicate the inputted token categories. However, in the case of LiLT, as there is no inclusion of image information, these two *Other embeddings* do not exist. Therefore, we employed the representation method shown in `Figure 2` to cover the structure of the multi-modal pre-trained models we use as extensively as possible.
> > >
> > > - **Question #5: Benchmark concerns**
> > >
> > >    Thank you for your comment. In the field of SER on visually-rich documents, FUNSD and CORD are widely recognized and frequently used benchmarks, while XFUND is a classic dataset for validating the multilingual capabilities of models. These datasets have been widely used in many existing models such as [LayoutLMv3](https://dl.acm.org/doi/abs/10.1145/3503161.3548112), [LayoutLMv2](https://aclanthology.org/2021.acl-long.201/), [LiLT](https://aclanthology.org/2022.acl-long.534/), [BROS](https://ojs.aaai.org/index.php/AAAI/article/view/21322), [Selfdoc](https://openaccess.thecvf.com/content/CVPR2021/html/Li_SelfDoc_Self-Supervised_Document_Representation_Learning_CVPR_2021_paper.html), and [DocFormer](https://openaccess.thecvf.com/content/ICCV2021/html/Appalaraju_DocFormer_End-to-End_Transformer_for_Document_Understanding_ICCV_2021_paper.html) to demonstrate their information extraction performance, making them broadly accepted in the research of SER on visually-rich documents. Moreover, our experimental results are based on an average of 10 trials, providing a robust evaluation of our findings. We acknowledge the limitations of our current approach due to language proficiency. We have only conducted experiments on the Chinese subset of XFUND. In future work, we plan to extend our testing to other languages included in XFUND, with the assistance of speakers of those languages. This will help us further explore the performance of our proposed method across different languages.
> > >
> > > - **Question #6: Example when run with PPTSER**
> > >
> > >    We appreciate your feedback and understand that providing more details about the tag-related prompt could help readers better understand our training process. Due to constraints in the main text, we have included the experimental protocol for our method in `Appendix A.2`. Furthermore, we provide an example of the process of constructing the tag-related prompt and training on the FUNSD dataset in `Appendix A.3`. We hope these additions will provide a clearer understanding of our method and its implementation. We value your suggestions and will consider including more detailed examples in future work.

---

> > > > ### Author Response · Authors · 2023-11-20
> > > > **Response to Reviewer 1jv5 (4/4)**
> > > >
> > > > ## Question
> > > >
> > > > - **Question #7: Explorations on CORD benchmark**
> > > >
> > > >    We appreciate your constructive question. In regards to the discrepancy in improvements between the FUNSD and CORD datasets, we have several speculations.
> > > >    - **Sample size:** Under the same K-shot setting, the CORD dataset contains a greater number of samples. The CORD dataset contains 30 entity types, and a single CORD sample cannot include entities of all types. Hence, the K-shot experiments demand more than K samples in the training set. This is in line with our description in `Section 3.1`. For instance, in the 1-shot experiment, our training set contained an average of 7.6 samples. However, this is still far less than $N \times K = 30$ samples, because one sample may contain entities of several types, and the support samples for different entity types might be the same. This is consistent with our descriptions in `Section 3.1` and `Appendix A.1`, where we chose as few samples as possible while covering the entity categories of CORD. In contrast, in the FUNSD dataset, a single sample usually includes entities of all 4 types, so there is only 1 sample in the training set in the 1-shot scenario. Our observations indicate that as the number of samples increases, the performance gap between our method and conventional fine-tuning decreases, yet our method maintains its advantage. Therefore, under the same K-shot setting, CORD involves more samples than FUNSD, resulting in less pronounced improvement.
> > > >
> > > >    - We conducted **additional experiments** with LayoutLMv2 and LiLT on the CORD dataset, and the results (shown in the table below) confirm our speculations.
> > > >
> > > >      | Sample Num | LayoutLMv2 FT | LayoutLMv2 Ours | LiLT FT | LiLT Ours |
> > > >      |:----------:|:-------------:|:---------------:|:-------:|:---------:|
> > > >      | 1          | 33.28         | **33.81**       | 40.46   | **42.47** |
> > > >      | 2          | 44.86         | **47.09**       | 49.39   | **54.18** |
> > > >      | 3          | 54.24         | **56.91**       | 56.31   | **63.71** |
> > > >      | 4          | 61.54         | **62.23**       | 61.36   | **69.19** |
> > > >      | 5          | 63.89         | **64.92**       | 64.09   | **69.83** |
> > > >
> > > >      As can be seen, when the number of samples we used was less than the number in the 1-shot setting (an average of 7.6 samples), the best improvement was better than that in the 1-shot setting, which also confirms our speculation.
> > > >    - **Label Complexity:** We believe that the labels in CORD are more complex and abstract, and the model's understanding of their semantic meanings is weaker than in FUNSD. For instance, the entity categories in FUNSD include *'header'*, *'question'*, and *'answer'*, while in CORD, they are more abstract types like *'menu.num'* and *'total.creditcardprice'*. Our supplementary experiments show that when the number of samples is extremely small, the improvement brought by our method is limited. However, this improvement increases rapidly with the number of samples, indicating that our method can more accurately grasp the relationship between document tokens and tag-related prompts with sufficient samples.
> > > >
> > > >    - **Pre-trained Models:** Different pre-trained models have varying degrees of understanding of labels. This could explain why some pre-trained models show weaker improvement on CORD. Our supplementary experiments reveal that the extent of improvement of our method is consistently better with the LiLT model than the LayoutLMv2 model, suggesting that LiLT better understands the semantic information implied by the labels.

---

### Official Review · Reviewer_Ueti · 2023-10-30

**Soundness:** 3 good
**Presentation:** 3 good
**Contribution:** 3 good
**Rating:** 8
**Confidence:** 4

**Summary:**

To address few-shot Semantic Entity Recognition (SER) in visually-rich documents, the authors introduce PPTSER, a pluggable approach to existing multimodal pre-trained models. PPTSER reframes SER into two sub-tasks: entity typing, which assigns entity types to each token in the document, and span detection, which determines whether tokens are at the beginning or middle of an entity span. The core of PPTSER involves (1) using SER tags as a prompt, concatenating them with the document’s tokens, and inputting them into a multimodal pre-trained model, and (2) using the attention weight from the last attention block between the tag-related prompt and the document’s tokens as the probability of tokens belonging to each tag. Consequently, PPTSER eliminates the need for a classifier layer, reducing the total number of parameters. Experimental results on widely used SER benchmarks demonstrate that PPTSER outperforms both traditional fine-tuning methods and few-shot methods in both few-shot and full-data scenarios. The authors also conduct additional analyses of PPTSER to validate its effectiveness.

**Strengths:**

PPTSER can be considered an original and significant contribution to the field. This is the first method that leverages cross-attention between tokens and tags to predict entities for SER tasks. Experimental results demonstrate that PPTSER significantly outperforms other models  in both few-shot and full-training-set scenarios. In addition, the model can be plugged to any pre-trained model, providing a versatile approach. The paper is overall clear and well-written.

**Weaknesses:**

Section 3.2 could be made clearer by using a more formal formulation of the model -- rather than giving the building blocks of the neural architecture. There are many complexities introduced because of this low level description.

In section 3.2 also, It would be good also to provide an example of an input sequence augmented with a tag-related prompt would improve comprehension.

No experimental details are given in section 4 (reproducibility issues).

**Questions:**

In section 3.2.1, could you clarify the last paragraph. Why the special "-1" is needed? It might be easier to just ignore the $\tilde c$ in the span detection loss.

In eq. (2), how are the matrices $Q$ and $K$ determined?

In section 4, the experimental protocol is not described (and no information in appendix on this side):

- Please give details (training details)
- In section 4.3, could you explain how PPTSER was trained?

In section 5 “Analysis”, could you specify the proportion of unrelated words added to the prompt?

---

> ### Author Response · Authors · 2023-11-20
> **Response to Reviewer Ueti**
>
> Thank you for your patient review. Our following responses will address your concerns.
> ## Weakness
> - **Weakness #1: Questions on the representation of figures and the formulations**
>
>    Thank you for your suggestion. Our original intent was to provide an intuitive understanding of our model's design through `Figure 2`, but we recognize that the representation might have inadvertently introduced some complexities. To address your concern, we have made modifications to `Figure 2`, removing certain non-essential elements to improve its clarity and intuitiveness. We will also polish the presentation to achieve a more formal and streamlined formulation of our model without sacrificing the necessary details of our neural architecture.
>
>    These updated figures and the refined formulation of our method will be included in the final versions of our paper. We hope these minor adjustments will enhance the readability and comprehension of our work.
>
> - **Weakness #2, Weakness #3, and Question #3: Implementation details**
>
>    Thank you for your suggestions. Due to space constraints in the main text, we have included additional experimental details in `Appendix A.2`, and provided examples to illustrate how our PPTSER is trained in `Appendix A.3` in our updated PDF files.
>
> ## Question
>
> - **Question #1: Questions on labels for span detection**
>
>    We apologize for the confusion caused. As you correctly pointed out, when the label for span detection ${y}^{det.}_i$ is set to -1, it implies that we ignore the loss for span detection at this position. We have already made this correction in the main text.
>
> - **Question #2: Questions on how ${Q}$ and ${K}$ are calculated**
>
>    We apologize for not explaining this clearly in the main text. ${Q}$ and ${K}$ correspond to *Query* and *Key* in the attention mechanism, respectively. When the hidden state from the second last layer ${H}^{t-1}$ is segmented into ${H}^{t-1}_i$ along the channel dimension, it is multiplied separately with learnable weight $({W}^{t}_i)_q$ and $({W}^{t}_i)_k$ to yield queries ${Q}^{t}_i$ as well as keys ${K}^{t}_i$ as follows:
>
>    $${Q}^t_i = ({W}^{t}_i)_q \times {H}^{t-1}_i$$
>
>    $${K}^t_i = ({W}^{t}_i)_k \times {H}^{t-1}_i$$
>
> - **Question #4: Replacement proportion of unrelated words and random embeddings**
>
>    In `Section 5`, under the subsection of `Prompt Engineering`, when we mention *unrelated words* or *random embeddings*, we refer to replacing all tag names in the prompt with unrelated words or random embeddings. For instance, replacing *menu.unitprice* in the prompt with *apple*, or replacing its embedding with a randomly initialized tensor, which can be considered as a 100\% replacement rate. This approach tests whether the semantic information contained in the tag names truly aids in completing the SER task. We apologize for any confusion caused by the text in `Section 5`. We will take steps to make this point clearer in the final version of the paper.
>
> Once again, we would like to express our gratitude for your diligent review and valuable feedback during the peer-review process.

---

### Official Review · Reviewer_GDwj · 2023-10-31

**Soundness:** 3 good
**Presentation:** 3 good
**Contribution:** 2 fair
**Rating:** 6
**Confidence:** 3

**Summary:**

This paper presents PPTSER, a few-shot method for semantic entity recognition on visually-rich documents. In PPTSER, SER tags are concatenated with the document tokens to serve as the input, and the class-wise logits are extracted from the last self-attention layers for few-shot learning and inference. The authors decouple the SER task into entity typing and span detection, and perform classifications on the two sub-tasks via extracting attention on two set of tags. PPTSER shows consistent improvements in few-shot settings on a range of datasets.

**Strengths:**

- PPTSER shows strong improvements in few-shot SER.
- The architecture can be applied into all kinds of transformer-based multi-modality model.
- The method's presentation is clear and well-structured, with a transparent design and motivations.
- The rationality is verified by careful analysis, strengthening the credibility of the proposed approach.

**Weaknesses:**

The paper acknowledges that using NER tags as prompts has been explored in text-based NER. This diminishes the novelty of the paper and raises concerns about its contribution in comparison to existing work.

**Questions:**

The paper mentions text-based few-shot NER frameworks in the Related Work. It would be valuable to clarify if these frameworks can be directly applied to SER on visually-rich documents with minimal or no significant modifications. If yes, a comparison with PPTSER would provide insights into its advantages and novelty.

---

> ### Author Response · Authors · 2023-11-20
> **Response to Reviewer GDwj**
>
> ## Weakness
> - **Weakness #1: Novelty Concerns**
>
>    I appreciate your feedback on the novelty of our work. While we agree that using NER tags as prompts has been explored in text-based NER, **the novelty of our work lies in the application to Semantic Entity Recognition (SER) tasks in visually-rich documents, a domain that has not been thoroughly studied yet**. The following points summarize our contributions:
>
>    - **Exploration of SER Tags' Effect on Visually-Rich Documents:** Unlike pure text pre-trained models such as BERT, the corpora used for pre-training multi-modal pre-trained models do not mimic natural language expression. Considering the fact that multi-modal pre-trained models are trained on structured documents whose corpora present a gap between the natural language expressions. This presents a new question of whether SER tags, which are closer to natural language expressions, can be understood by multi-modal pre-trained models and guide SER tasks. Our experiments have successfully validated the effectiveness of the proposed approach.
>
>    - **Adaptation for Visually-Rich Documents:** We have shown that text-based few-shot NER methods may not be as effective for the SER task on visually-rich documents. However, Our method, which is adapted for entity-rich scenarios on visually-rich documents, outperforms these conventional methods, highlighting the significance of this research.
>
>    - **Efficient Use of Multi-Head Attention Mechanism:** Our method extends beyond using SER tags as prompts by leveraging the multi-head attention mechanism within the Transformer architecture. Contrary to other methods that utilize cross-attention, we did not employ an additional cross-attention module. Instead, we utilized the self-attention blocks within the multi-modal pre-trained model to perform the functions of cross-attention. This approach saves memory and computational time. Additionally, we leveraged the multi-head attention mechanism inherent in the self-attention block, which allows different attention heads to focus on varied content, offering mutual backup and reducing the incidence of missed recognition of entities.
>
>    - **Entity Classification and Entity Boundary Detection:** Our work divided the task of SER on visually-rich documents into two sub-tasks: entity typing and span detection, and addressed both using the same framework. It has successfully addressed the task of entity extraction under the BIO tagging scheme that fits visually-rich documents. We have innovatively used natural language descriptions as prompts (*beginning* and *inner*) for span detection. In the context of NER on plain text, the demarcation of boundaries between entities has not been sufficiently emphasized. However, in SER tasks on visually-rich documents, distinguishing the boundaries of entities accurately with the BIO tagging scheme is crucial due to the possibility of two entities of the same entity type being positioned adjacently. If their boundaries are not correctly distinguished, they will be erroneously merged into a single entity.
>
>    In conclusion, our work presents a novel approach to few-shot SER tasks on visually-rich documents, addressing unique challenges that have not been explored in existing plain text-based NER methods. We believe that these contributions significantly extend the existing body of knowledge on entity recognition tasks.
>
> ## Question
> - **Question #1: Modifications on few-shot NER methods**
>    - Thank you for your suggestion. In `Section 2`, only a few methods we mentioned share the same In-Label-Space setting as ours, including **COPNER** and **EntLM**. These methods hold potential for applications on visually-rich documents. We have experimented with these methods, and the results have been included in `Table 2` of our paper. The performance of these methods was found to be less satisfactory compared to ours, affirming the effectiveness of our proposed method. Given the constraints on the length of the main text, we initially did not provide detailed information about the modifications made to these comparison methods. However, to address your concerns, we have now updated our document and included the details of the modifications in `Appendix A.2` of our paper.

---

> > ### Comment · Reviewer_GDwj · 2023-11-23
> >
> > Thanks for your response. I acknowledge that I've read it.

---

### Meta-Review · Area_Chair_7CPz · 2023-12-12

**Metareview:**

After careful consideration of the reviews and the authors' rebuttal, I lean towards a rejection decision on the paper. The core concerns pointed out by the reviewers relate to the novelty and real-world applicability of the proposed approach. While some reviewers recognized the strengths in experimental performance and the architectural clarity, there exist concerns about the novelty and label leakage issue, coupled with a lack of significant improvement on the CORD dataset, remain substantial.

The authors have made efforts to address the reviewers' concerns, arguing the uniqueness of their approach and its suitability for few-shot SER tasks with visually-rich documents. However, the rebuttal has not sufficiently alleviated the concerns regarding the method's applicability to real-world scenarios and its distinction from prior work in the field. The fact that the method's performance does not significantly improve beyond certain experimental setups raises questions about its scalability and effectiveness in diverse settings.

Considering the thorough and constructive feedback from the reviewers, alongside thoughtful responses from the authors, it is concluded that while the paper presents interesting ideas and notable experimental outcomes, the issues around novelty, real-world applicability, and inconsistent performance across datasets highlight the need for further work before the research can be considered for acceptance.

**Justification For Why Not Higher Score:**

While some reviewers recognized the strengths in experimental performance and the architectural clarity, there exist concerns about the novelty and label leakage issue, coupled with a lack of significant improvement on the CORD dataset, remain substantial.

**Justification For Why Not Lower Score:**

N/A

---

### Decision · Program_Chairs · 2024-01-16

Reject